

# 1 Tree proximity affects soil respiration dynamics in a

# 2 coastal temperate deciduous forest

Stephanie C. Pennington*[1], Nate G. McDowell[2], J. Patrick Megonigal[3], James C. Stegen[2], and
Ben Bond-Lamberty[1]
*Corresponding author, stephanie.pennington@pnnl.gov
1. Joint Global Change Research Institute, Pacific Northwest National Laboratory, 5825
University Research Ct. #3500, College Park, MD 20740 USA
2. Pacific Northwest National Laboratory, Biological Sciences Division, Richland, WA, USA
3. Smithsonian Environmental Research Center, Edgewater, MD, USA
Keywords: *spatial variability, soil respiration, temperate forest, carbon cycling*

**18 Abstract**

Soil respiration ($R_s$), the flow of $CO_2$ from the soil surface to the atmosphere, is one of the
largest carbon fluxes in the terrestrial biosphere. The spatial variability of $R_s$ is both large and
poorly understood, limiting our ability to robustly scale it in time and space. One factor in $R_s$
spatial variability is the autotrophic contribution from plant roots, but it is uncertain how the
proximity of plants affects the magnitude and temperature sensitivity of $R_s$. This study examined
the effect of tree proximity on $R_s$ in the growing and dormant seasons, as well as during





moisture-limited times, in a temperate, coastal, deciduous forest in eastern Maryland, USA. In a
linear mixed-effects model, tree basal area within 5 m ($BA_5$) exerted a significant positive effect
on the temperature sensitivity of soil respiration. Soil moisture was the dominant control on $R_S$
during the dry portions of the year while soil moisture, temperature, and $BA_5$ all exerted
significant effects on $R_S$ in wetter periods. Our results suggest that autotrophic respiration is
more sensitive to temperature than heterotrophic respiration at these sites, although we did not
measure these source fluxes directly, and that soil respiration is highly moisture-sensitive, even
in a record-rainfall year. The $R_S$ flux magnitudes (0.3-16.6 µmol m$^{-2}$ s$^{-1}$) and variability
(coefficient of variability 10%-22% across plots) observed in this study were comparable to
values observed over decades in similar forests. We estimate that four $R_S$ observations were
required to be within 50% of the stand-level mean, and 311 to be within 5%, at 90% confidence.
A better understanding of the spatial interactions between plants and microbes that results in
measured $R_S$ is necessary to link these processes with large scale soil-to-atmosphere C fluxes.

**Introduction**
Soil respiration ($R_s$), the flow of $CO_2$ from the soil to the atmosphere, is an important
carbon (C) flux at ecosystem (Granier et al., 2000) to global scales. $R_s$ is among the largest C
fluxes in the terrestrial biosphere (Bond-Lamberty, 2018; Le Quéré et al., 2018), but poorly
constrained at large scales, and thus it is important to understand its variability and sensitivity to
processes such as land use and climate changes (Hursh et al., 2017; Schlesinger and Andrews,
2000). Unlike other large C fluxes such as net primary production, net ecosystem exchange,
and gross primary production, $R_s$ cannot be measured, even indirectly, at scales larger than ~1
m$^2$ (Bond-Lamberty et al., 2016), limiting our ability to robustly scale it in time and space.
One obstacle to robust measurements is that the spatial variability of $R_s$ is both large
and poorly understood. Controls on the spatial variability of $R_s$ differ among sites and
ecosystems and include plant species, leaf habit, ecosystem productivity (Reichstein et al.,



2003), soil temperature, moisture, spatial variability of vegetation, management, and soil
compaction (Epron et al., 2004). This high variability has consequences for the sampling
strategy required to accurately measure $R_s$ at the stand scale (Rodeghiero and Cescatti, 2008;
Saiz et al., 2006) and limits our ability to upscale $R_s$ measurements to eddy covariance tower
scales (Barba et al., 2018).

At large scales, $R_s$ differs between vegetation types and biomes (Raich et al., 2002;

Raich and Schlesinger, 1992), implying that the spatial distribution of vegetation might strongly
affect $R_s$ via plant root respiration, which constitutes ~50% of $R_s$ in many ecosystems (Subke et
al., 2006). At ecosystem scales, a number of studies have examined how the spatial distribution
of $R_s$ is affected by vegetation. $R_s$ is typically higher closer to tree stems (Epron et al., 2004;
Tang and Baldocchi, 2005), and with higher nearby stem density (Stegen et al., 2017).
Photosynthesis is also a driver of the rhizospheric component of soil respiration (Hopkins et al.,
2013), and influences seasonal trends in root contribution to total soil respiration (Brændholt et
al., 2018; Högberg et al., 2001). Any spatial influences of plants on $R_s$ might be expected to be
particularly strong in temperate, deciduous forests, as such forests tend to be especially
productive (Gillman et al., 2015; Luyssaert et al., 2007).

This study examines the effect of tree proximity on measured $R_s$ in a mid-Atlantic,

deciduous forest in the Chesapeake Bay, USA region. We hypothesized that:

(i) the amount of basal area close to $R_s$ measurement locations would exert a significant and
positive effect on measured $R_s$ after taking into account the effects of abiotic drivers;

(ii) this effect would occur in the growing (leaf on) season, but not in the dormant (leaf off)
season, because root respiration is much stronger during the growing season; and



(iii) this effect would be stronger during drier times of year, because trees might maintain access
to deep soil moisture (Burgess et al., 1998) and thus continue respiring even when the surface
soil is dry.

To test these hypotheses we performed a spatially explicit analysis of one year of frequent $R_s$
measurements in a temperate coastal deciduous forest in eastern Maryland, USA. To our
knowledge, no study has examined the influences of trees on spatial variation of $R_s$ in the
Chesapeake Bay watershed, an area subject to rapid rates of sea level rise (Ezer and Corlett,
2012; Sallenger et al., 2012) that may exert significant effects on the carbon cycling of coastal
ecosystems (Rogers et al., 2019).

**Methods**

*Site characteristics*

This study was conducted in a mid-Atlantic, temperate, deciduous forest at the

Smithsonian Environmental Research Center (SERC) in Edgewater, MD, USA. Three sites were
chosen along Muddy Creek, a stream draining into an arm of Chesapeake Bay. Each site was
separated by ~1 km (**Figure 1a**). These sites were comprised of both lowland and upland forest
with a mean annual precipitation of 1001 mm and mean annual temperature of 12.9°C (Pitz and
Megonigal, 2017). Dominant tree species include *Liriodendron tulipifera*, *Fagus grandifolia*, and
*Quercus spp.*; soil types vary between Collington, Wist, and Annapolis soil. (**Table 1**). At each
site, three 20 m x 40 m plots were installed, separated by ~25 m and oriented perpendicular to
the creek. The total elevation change between plots at each site was ~2 m. Within each plot, we
installed 4, 20-cm diameter PVC collars, randomly separated from each other by 2–15 m, for a
total of 36 measurement collars. Collars were installed ~1 week prior to the first sampling and
left in place for the duration of the study.




*Soil respiration measurements*

Soil respiration measurements were taken using an infrared gas analyzer (LI-8100A, LI-

COR Inc., Lincoln, NE) with a 20 cm diameter soil chamber attached. Measurements were
taken every 10-14 days from April 2018 to April 2019. The IRGA measures concentrations every
second over a one minute period and calculates the $CO_2$ flux as the linear or exponential
regression of $CO_2$ accumulation in the closed chamber system over unit area and time; two
successive measurements were taken at each collar and averaged. Vegetation was removed
from inside the collar, and new vegetation was re-clipped as necessary, to remove any
aboveground autotrophic flux, so that the IRGA was measuring only soil-to-atmosphere $CO_2$.
Soil moisture and temperature ($T_5$) were also recorded at 5 cm depth, using auxiliary sensors
attached to the LI-8100A, at the same time as soil respiration measurements. Temperature at
20 cm depth ($T_{20}$) was also recorded using a hand-held thermometer at the time of
measurement.

*Tree proximity measurements*

We recorded distance from the soil collar, diameter at breast height (1.37 m), and

species of each tree within a 15 meter radius of each soil respiration measurement point
(**Figure 1b**).  Dead trees were included in the dataset but only account for < 1% of total forest
basal area. Cumulative basal area was calculated at each 1 m radial distance from the collar,
summing the cross-sectional areas of all trees within each distance. Tree root extent can be
highly variable, but generally roots extend at least to the edge of the tree canopy (Stone and
Kalisz, 1991). Mature tree canopies at SERC are ~5 m in radius (S. Pennington, personal
observation), and we adopted this distance as an *a priori* assumption to test for the effect of
basal area at 5 meters ($BA_5$) on $R_s$.



*Statistical analysis*

Respiration data were checked visually for artifacts or unusual outliers, but we did not

exclude any data *a priori*. Data were then combined with the proximity measurements described
above based on collar number. We used a linear mixed-effects model to test for the influence of
$BA_5$ on $R_s$, treating temperature, soil moisture, $BA_5$ as fixed effects, and site as a random effect.
To ensure homoschedasticity of model residuals, the dependent variable $R_s$ was transformed by
taking its natural logarithm. We used restricted maximum likelihood estimation using the *lme4*
package (Bates et al., 2015) in R version 3.5.3 (R Development Core Team, 2019). All models
were examined for influential outliers and deviations from normality. Non-significant terms were
then eliminated using a forward-and-back stepwise algorithm (using the R package *MASS*
version 7.3-47) based on the Akaike Information Criterion. Residuals from all fitted models were
plotted and checked for trends or heteroschedasticity.

Our secondary hypotheses, that effect of $BA_5$ varies with growing season and soil

moisture, were tested by subsetting the $R_s$ data. We treated April 15-October 14 as the growing
season, based on 2018 leaf-out and senescence, and October 15-April 14 as the dormant
season. Soil moisture data were split up into equal thirds (low, $<0.188$ $m^3$ $m^{-3}$; medium, 0.188-
0.368 $m^3$ $m^{-3}$; and high, $>0.368$ $m^3$ $m^{-3}$; all values volumetric). We then applied the statistical
model described above to each subset of the data.

We used the spatial variability between collars within individual plots to estimate the

number of samples required for a robust estimate of the $R_s$ 'population mean', i.e., a spatially-
representative mean. Specifically, we used a Student's t-test to calculate this based on the
standard deviation of hourly $R_s$, the desired power of the test, and the allowable delta
(difference from the true mean value), following Davidson et al. (2002).

All code and data necessary to reproduce our results are available in our online GitHub

repository (https://github.com/PNNL-PREMIS/PREMIS-ghg) and permanently archived at
Figshare (DOI if accepted).





**Results**

We measured $R_s$, soil temperature, and soil moisture on 31 different days across the
one-year period (Figure 2). Soil temperatures ranged from 0.1 to 27.7 ℃ (at 5 cm) and 1.7 to
24.4 ℃ (at 20 cm); volumetric soil moisture values were 0.01-0.56. $R_s$ fluxes ranged from 0.17
$\mu$mol m$^{-2}$ s$^{-1}$ (in March 2019) to 16.55 $\mu$mol m$^{-2}$ s$^{-1}$ (in July 2018). The coefficient of variability
(CV) between collars within plots, a measure of spatial variability, ranged from 10% to 22%.
This implied that a large number of samples was required to estimate soil respiration accurately
(Table 2).
There was large variability in the basal area and number of trees close to the
measurement collars (Figure 3). The mean number of trees within 1 m, 5 m, and 10 m distance
were one, six, and 20 trees (with respective nearby basal areas of 0.0002 m$^2$, 0.24 m$^2$, and 0.91
m$^2$). Within our maximum radius of measurement, 15 m, there were on average 42 trees and 1.7
m$^2$ of cumulative basal area, ranging from a minimum of 0.55 m$^2$ to a maximum of 3.55 m$^2$. The
forest was thus highly spatially variable in its distribution of trees relative to the $R_s$ measurement
collars.

*Effect of BA on $R_s$*
The linear mixed-effects model using temperature, soil moisture, and basal area within
five meters (BA$_5$) predicted almost half of the $R_s$ variability (conditional $R^2$ = 0.40). BA$_5$ was not
significant by itself in a Type III ANOVA using this model ($\chi^2$ = 0.495, P = 0.482), but exhibited
strong and significant interactions with $T_5$ and $T_{20}$ (**Table 3**). In addition, the residuals of a model
fit without BA$_5$ had a significant trend with BA$_5$ (**Figure 4**). Separating the data into growing- and
dormant-season subsets provided contrasting results. In the growing season, model outputs
were similar to those of the full year model, with BA$_5$ having significant interactions with $T_5$ and



$T_{20}$ (data not shown). The dormant season model, however, was quite different: only $T_{20}$ (P ≤
0.001) and soil moisture (P = 0.0009) were significant terms. In addition, the dormant season
model explained more of the $R_s$ variability (AIC = 258.75, marginal $R^2$ = 0.52). In summary,
collars with higher basal area within 5 m had significantly higher temperature sensitivity of soil
respiration after controlling for temperature and moisture effects, while basal area within 5 m of
sampling points was not correlated with $R_s$ during the dormant season.

Our third hypothesis was that any basal area effect on $R_s$ would be strongest in the

driest times of the year, when microbial respiration at the surface soil declines as the soil dries,
but (we speculated) trees would maintain access to deeper soil moisture. There were in fact
strong differences between the driest and wettest thirds of the data, but our hypothesis was not
supported. In the driest third of the data, neither $BA_5$ nor its interaction with $T_5$ was significant (P
= 0.1775 and 0.1078 respectively); $T_{20}$ was never significant; and the dominant control was
instead soil moisture ($\chi^2$ = 20.93, P < 0.001). In contrast, the wettest-third model resembled the
full-year model, with $BA_5$ interacting with temperature, and soil moisture also significant.

*Sensitivity test*

Our *a priori* choice of 5 m for the basal area test was one of many possible choices, and

could potentially bias the results, as the actual extent of tree roots at these sites is unknown.
Re-running the main statistical test across a wide range of distances, however, showed that
basal area by itself was almost never significant, while its interactions with $T_5$ and $T_{20}$ were
almost always significant (**Figure 5**). Generally the BA effects were not significant at short (< 3
m) distances; this is expected, given that few collars were that close to trees. Interestingly, the
BA effects remained significant all the way to our maximum measured distance of 15 m. In
summary, our analytical choice of a 5 m radius did not appear to bias our results.




**Discussion**

*Results and implications of $R_s$ values*
The $R_s$ fluxes observed in this study, 0.3-16.6 µmol m$^{-2}$ s$^{-1}$, were comparable to values in
similar forests (Giasson et al., 2013) and from the Soil Respiration Database (Bond-Lamberty
and Thomson, 2010), a synthesis of annual $R_s$ studies (0 to 14.7 µmol m$^{-2}$ s$^{-1}$, n = 1281
temperate deciduous studies). We observed a collar-to-collar $R_s$ CV of 10.5-21.5%, a value also
comparable to previous studies. In a study of $R_s$ in conifer forests and grasslands, Rodeghiero
(2008) reported 28.9-41.5% variability, Davidson et al. (2000) about 30% in forest ecosystems,
and a much broader range (0.11-84.5%) for temperature, deciduous forests from the SRDB.
Sample size requirements to estimate annual $R_s$ were high at SERC compared to
previous studies. For example, to be within 10% of the mean $R_s$ flux at 95% confidence required
from 41 (Davidson et al., 2002) in Harvard Forest, to 72 (Adachi et al., 2005) in a secondary
forest, to 133 sample points in this study. This high variability between studies likely arises
because controls on the spatial variability of $R_s$ differ among sites and ecosystems. Within forest
biomes, topography and stand structure (Søe and Buchmann, 2005) can also be dominant
controls that likely contribute to the high variability seen in this study.

*Interactions between basal area and temperature sensitivity on $R_s$*
Many studies have examined whether autotrophic respiration ($R_a$) or heterotrophic
respiration ($R_h$) is more temperature-sensitive, and reached varying conclusions (Aguilos et al.,
2011; Boone et al., 1998; Wang et al., 2010). In this study, however, collars with higher basal
area within 5 m had significantly higher temperature sensitivity of soil respiration after controlling
for temperature and moisture effects. This suggests that $R_a$ is more sensitive to temperature
than $R_h$ at these sites, even though we did not directly measure the autotrophic and
heterotrophic source fluxes contributing to the overall $R_s$ flux.





Mechanistically, these findings could be explained by a number of processes.  For

example, when substrate supply from root exudates is ample, $R_s$ tends to be more sensitive to
temperature (Luo and Zhou, 2006), presumably because $R_s$ can be tightly coupled with
photosynthesis and thus roots, which access the photosynthate before microbes, respond more
strongly to temperature changes. There is also abundant evidence that soil moisture influences
temperature sensitivity: Suseela et al. (2012), for example, found that $R_s$ is less sensitive to
temperature during water-limited times. If trees' roots have access to water consistently, their
respiratory flux $R_a$ measured at the soil surface as part of $R_s$ will be more temperature-sensitive
on average, because $R_a$ will be limited by soil moisture less frequently (Misson et al., 2006). It is
important to note that these various mechanisms are not mutually exclusive.

*Soil moisture controls on BA significance*

We hypothesized that $BA_5$ effect would be particularly strong during the driest third of the

year, but found that only soil moisture controlled $R_s$ during these periods, while neither
temperature nor tree proximity ($BA_5$) was significant. This demonstrates that $R_s$ is highly
moisture-sensitive at these sites, but does not support our hypothesis that trees might have
access to deeper or different water sources than surface soil microbes. Soil moisture is
considered to be a primary $R_s$ control in Mediterranean and desert ecosystems (Cable et al.,
2010), but interestingly even this deciduous forest, in a year with record rainfall (National
Weather Service, 2019), experienced significant moisture restrictions on $R_s$. Spatial variation in
soil moisture (CV 2.5%-18.7% between plots) was probably due to the topographic variability of
our study site, which allowed some measurement points to drain more quickly than others,
producing a wide range of soil moisture conditions.

*Dormant season $R_s$ controls*





Tree basal area within 5 m of our $R_s$ sampling points was not significant in the dormant
season model, supporting our hypothesis that total $R_a$ contribution is often lower during the
dormant reason than the growing season (Hanson et al., 2000), which suggests that $R_a$
contributes less to $R_s$ during the dormant season. This is expected, given the physiological link
between photosynthesis and root respiration (Sprugel et al., 1995). Interestingly, $T_5$ was not
significant in the dormant season model, but rather $T_{20}$ was the dominant control. The study site
is in a mid-Atlantic, temperate location with cold air temperatures during the winter. Deeper soils
are more insulated from cold air temperatures, allowing more favorable conditions for $R_s$ and
potentially making $T_{20}$ a dominant control during these times.

*Limitations of this study*
A number of limitations should be noted in our study design and execution. First, this
was not a fully spatially-explicit analysis; we did not map the collars relative to each other, nor
construct a full spatial map of the forest stands (Atkins et al., 2018). Such mapping can be
useful to examine the $R_s$ spatial structure in more detail, as for example in Stegen et al. (2017),
but our approach to mapping relative distances to trees provides an alternative spatial study
construct. In a similar vein, Tang and Baldocchi (2005) measured $R_s$ within a transect of two oak
trees to draw inferences on the spatially variable contribution of $R_h$ and $R_a$. This study design
still provides useful spatial information, however: the 15 m max distance in **Figure 5** implies that
the range of a semivariogram, i.e. the distance of maximum autocorrelation, would be at least
this far. This means that BA remained significant all the way to our maximum measured
distance of 15 m, implying that the spatial influence of large trees persisted at least this far
(Högberg et al., 2001).

This study tested the effect of basal area on $R_s$, based on the assumption that BA is
proportional to fine root biomass, the respiration of which is driven (with some time lag) by





photosynthesis and this in turn drives root respiration dynamics (Vose and Ryan, 2002). Stems
with a diameter below 2 cm and understory were not inventoried or, as a result, included in the
hypothesis-testing statistical models. If root respiration is instead correlated with number of
stems, which are disproportionately small due to forest demographics, this would bias our
results. There are not many understory/saplings at these sites (**Table 1**), however.

**Conclusion**
Autotrophic respiration was found to be more sensitive to temperature than heterotrophic
respiration, and collars with higher basal area within 5 m had significantly higher temperature
sensitivity. $R_s$ is also highly moisture-sensitive at these sites, with large differences among $R_s$
controls in low- versus high-moisture times. These findings, in conjunction with large sample
size requirements, suggest soil respiration at this site to be highly dynamic and variable. This
could have implications for measurement requirements in sites with particular stand structures.
A better understanding of the spatial interactions between plants and microbes that results in
measured $R_s$ is necessary to link these processes with collar- and ecosystem-scale soil-to-
atmosphere C fluxes.

**Acknowledgments**
This research is part of the PREMIS Initiative at Pacific Northwest National Laboratory (PNNL).
It was conducted under the Laboratory Directed Research and Development Program at PNNL,
a multi-program national laboratory operated by Battelle for the U.S. Department of Energy
under Contract DE-AC05-76RL01830.This research was supported by the Smithsonian
Environmental Research Center. We thank Alexey Shiklomanov for pointing out a crucial
mistake in our statistical code.

**Author contributions**



This study was designed by B.B.-L. and S.C.P. All fieldwork and data analysis was performed
by S.C.P., except for the statistical analysis, which was written by B.B.-L. N.M., J.P.M., and
J.C.S. provided feedback on the study design, analysis, and interpretation of results. S.C.P.
wrote the manuscript in close collaboration with all authors.

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





**Table 1** | Study site characteristics of each site along Muddy Creek, including trees per hectare,
cumulative basal area, main soil types, and dominant tree species by percent of basal area.
Values are mean ± standard deviation of N=3, 800 m$^2$ plots.

| Site | Trees (ha$^{-1}$) | BA (m$^2$ ha$^{-1}$) | Dominant Soil Type | Dominant Tree Species (by BA %) |
|---|---|---|---|---|
| GCReW (38.876 °N, 76.553 °W) | 637.5 ± 57.3 | 44.6 ± 4 | Collington-Wist complex; Collington and Annapolis soils | 28% *Liriodendron tulipifera* 11% *Quercus spp.* 11% *Fagus grandifolia* |
| Canoe Shed (38.884 °N, 76.557 °W) | 529.2 ± 93.8 | 40.4 ± 6 | Annapolis fine sandy loam | 26% *Quercus spp.*, 23% *L. tulipifera* 20% *F. grandifolia* |
| North Branch (38.887 °N, 76.563 °W) | 806.9 ± 180.7 | 34.5 ± 7.8 | Collington and Annapolis soils; Collington, Wist, and Westphalia soils | 42% *F. grandifolia* 26% *Quercus spp.* 12% *Liquidambar styraciflua* |







**Table 2.** Sample size required to estimate soil respiration with a particular error (delta, left
column, fraction of mean flux), for different statistical power values. Values are mean ± standard
deviation between plots. "Power" is the probability that the test rejects the null hypothesis when
a specific alternative hypothesis is true, and informally connotes the degree of confidence that
the measurement within some delta value of the true mean.

| | Power ($1 - \beta$) | | | | | |
|---|---|---|---|---|---|---|
| **Delta** | **0.5** | **0.6** | **0.7** | **0.8** | **0.9** | **0.95** |
| 0.05 | 63 ± 21 | 97 ± 33 | 147 ± 50 | 226 ± 76 | 373 ± 124 | 532 ± 175 |
| 0.10 | 16 ± 6 | 25 ± 9 | 37 ± 13 | 57 ± 19 | 94 ± 31 | 133 ± 44 |
| 0.25 | 3 ± 1 | 4 ± 2 | 6 ± 2 | 10 ± 4 | 15 ± 5 | 22 ± 7 |
| 0.50 | 1 ± 1 | 1 ± 1 | 2 ± 1 | 3 ± 1 | 4 ± 2 | 6 ± 2 |








**Table 3.** Summary of linear mixed-effects model testing main hypothesis of the effect of nearby
tree basal area on soil respiration (the dependent variable). Terms tested include soil
temperature at 5 and 20 cm ($T_5$ and $T_{20}$ respectively), basal area (BA), and soil moisture (SM).
Model AIC = 662.7, marginal $R^2$ = 0.72.

| | Value | Std.Error | DF | t-value | p-value |
|---|---|---|---|---|---|
| **(Intercept)** | -0.7824 | 0.1215 | 884 | -6.4418 | 0.0000 |
| **$T_5$** | 0.0146 | 0.0080 | 884 | 1.8327 | 0.0672 |
| **BA** | -0.1162 | 0.1659 | 884 | -0.7006 | 0.4837 |
| **$T_{20}$** | 0.0873 | 0.0093 | 884 | 9.3562 | 0.0000 |
| **SM** | 3.3107 | 0.5627 | 884 | 5.8834 | 0.0000 |
| **$SM^2$** | -5.4007 | 0.8867 | 884 | -6.0913 | 0.0000 |
| **$T_5$:BA** | 0.1165 | 0.0297 | 884 | 3.9144 | 0.0001 |
| **BA:$T_{20}$** | -0.1018 | 0.0332 | 884 | -3.0667 | 0.0022 |






**Figure 1** | a) Tree proximity measurement schematic. Distance to each tree was recorded within
a 15 meter radius of each soil respiration measurement point, along with DBH and species. b)
Map of the Smithsonian Environmental Research Center with the three sites labeled in black.

a)







b)

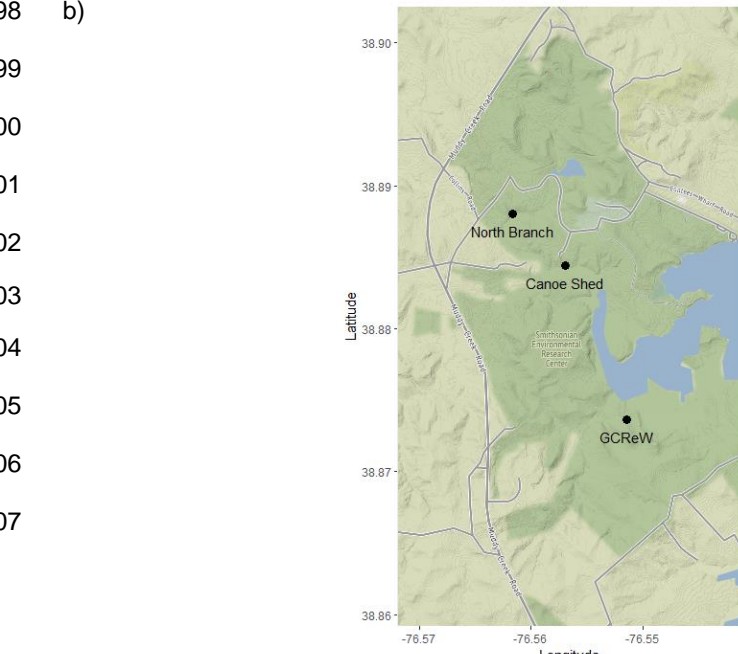














**Figure 2** | Mean flux over time from April 2018 to April 2019 for 36 measurement points across
three sites; blue line shows the seasonal trend using a loess smoother.

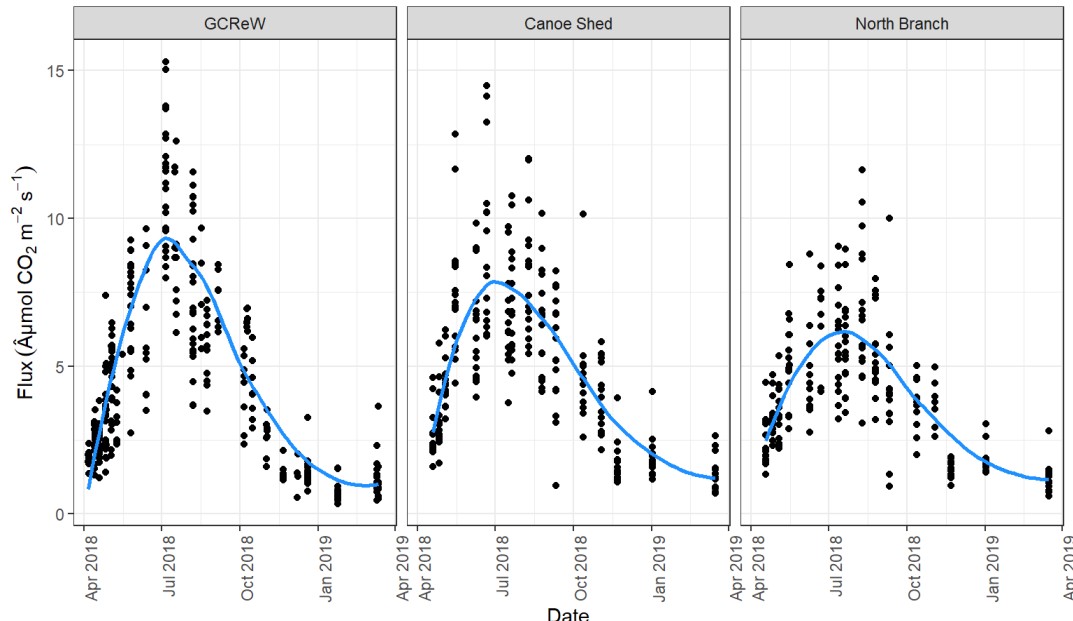






**Figure 3** | Cumulative basal area for each collar (N = 36) up to 15 meters; color indicates
number of trees at each distance.

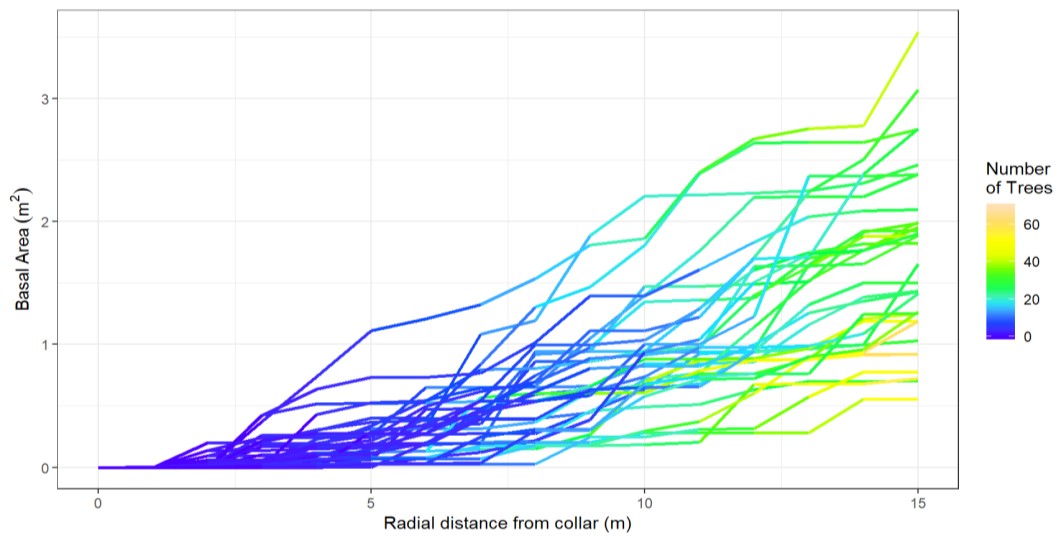







**Figure 4.** Residuals of a soil respiration model, incorporating temperature and soil moisture as independent variables, versus cumulative tree basal area within 5 m, by site. Each point is an individual observation (cf. Figure 2). Regression lines are shown for each site; black line is the overall trend.

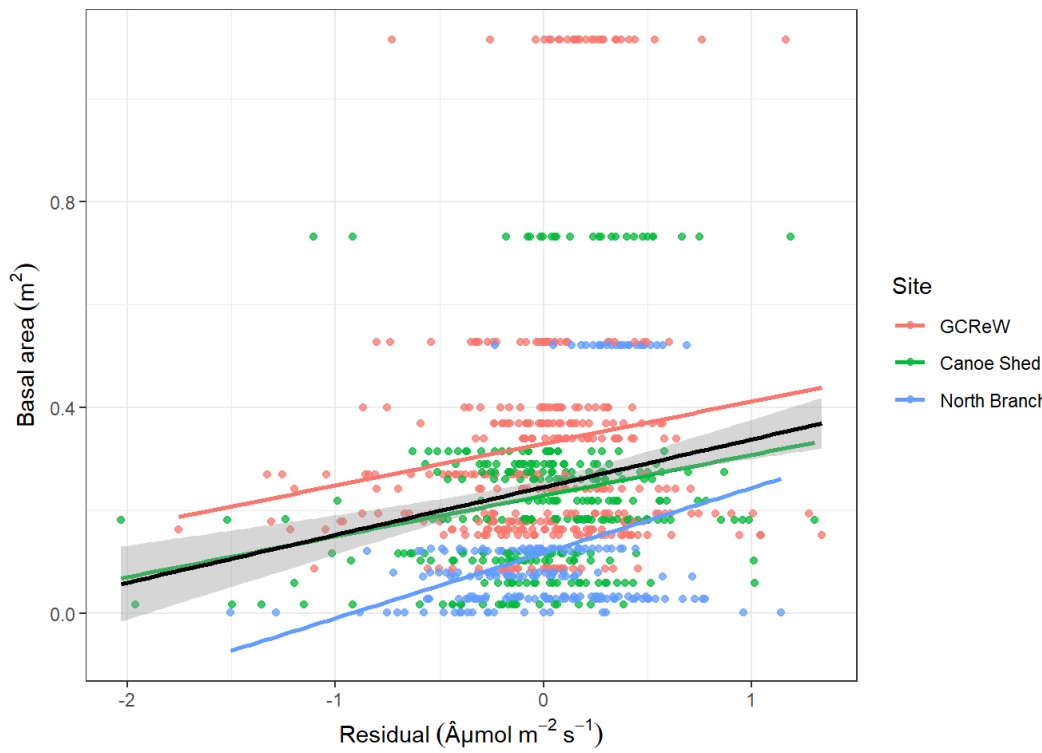



**Figure 5.** Test of robustness of results, run at various distances from measurement collars (x
axis). Figures shows the significance (chi square p-value from Type III ANOVA of the linear
mixed effects model, y axis; note logarithmic scale) of basal area (BA), as well as the interaction
of BA and temperatures at 5 and 20 cm ($T_5$ and $T_{20}$ respectively). Horizontal dashed line shows
the standard 0.05 significance cutoff; vertical dashed line the 5 m radius used in **Table 3** and
**Figure 4** results. Note that 'missing' green and blue dots at distances < 5 m mean that the
terms were dropped from the model and are thus not significant.

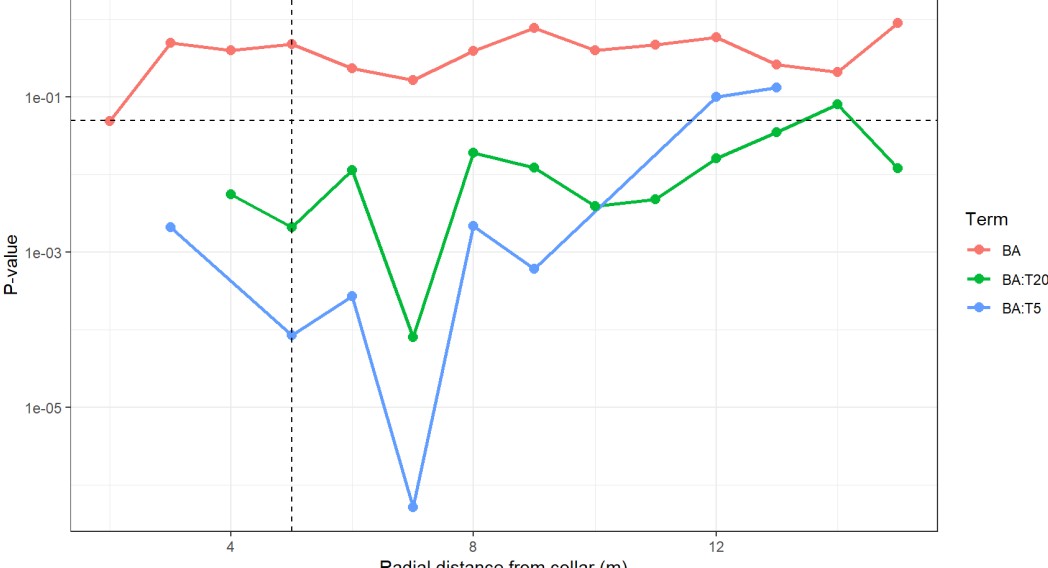






