# Peer review of "Localized basal area affects soil respiration temperature sensitivity in a"

_Biogeosciences, 2019_

## Referee Comment (RC1) · Anonymous Referee #1 · 24 Jul 2019

The manuscript entitled "Tree proximity affects soil respiration dynamics in a coastal temperate deciduous forest" is in fact addressing three different questions: (1) tree proximity and soil respiration, (2) temperature sensitivity, and (3) required sampling effort. Only the first one is clearly reflected in the title. These three questions are relevant and within the scope of BG, but they are not novel and there is no novel concept, idea or tool that emerged for this study. This is an additional set of data (a case study). (1) The approach of linearly connecting the basal area of trees to a fixed distance (5 m) and ground respiration is simplistic. The distance at which an individual tree influenced soil respiration is probably dependent on the size of this tree. In other words, biggest trees are expected to have a stronger influence than smaller trees. There are several

(many papers) relating addressing the effect of tree size and proximity on soil respiration that are not cited in this manuscript. Among them: Fang C, Moncrieff JB, Gholz HL, Clark KL (1998) Soil CO2 efflux and its spatial variation in a Florida slash pine plantation. Plant Soil 205:135–146. doi:10.1023/A:1004304309827 Metcalfe DB, Meir P, Aragão LEOC, Malhi Y, da Costa ACL, Braga A, Gonçalves PHL, de Athaydes J, de Almeida SS, Williams M (2007) Factors controlling spatio-temporal variation in carbon dioxide efflux from surface litter, roots, and soil organic matter at four rain forest sites in the eastern Amazon: PARTITIONING AMAZON SOIL RESPIRATION. J Geophys Res 112. doi:10.1029/2007JG000443 Katayama A, Kume T, Komatsu H, Ohashi M, Nakagawa M, Yamashita M, Otsuki K, Suzuki M, Kumagai T (2009) Effect of forest structure on the spatial variation in soil respiration in a Bornean tropical rainforest. Agric For Meteorol 149:1666–1673. doi :10.1016/j.agrformet.2009.05.007 Bréchet L, Ponton S, Alméras T, Bonal D, Epron D (2011) Does spatial distribution of tree size account for spatial variation in soil respiration in a tropical forest? Plant and Soil 347:293–303. doi: 10.1007/s11104-011-0848-1 Schwendenmann L, Macinnis-Ng C (2016) Soil CO2 efflux in an old-growth southern conifer forest (Agathis australis) – magnitude, components and controls. SOIL 2:403–419. doi: 10.5194/soil-2-403-2016 Reading these papers (but the list is not limitative) would have given way to analyze more finely the results, especially the last two.

(2) The observation that autotrophic respiration is more sensitive to temperature than heterotrophic respiration is also confirmative of many studies. Note that the paper Aguilos et al 2011 that is cited when discussing this point has not been accepted for publication in Biogeoscience, so the citation is wrong. Note that the citation Wei et al is incorrect: should be Wei et al (doi: 10.1016/j.soilbio.2010.04.013). The discussion of this fact is rather poor and miss one of the most important drivers of the apparent temperature sensitivity of RA: phenology. This may be important in the present study since soil respiration was measured over a full year and species are deciduous. Among many other sources, this has been discussed in: Epron D, Le Dantec V, Dufrêne E, Granier A (2001) Seasonal dynamics of soil carbon dioxide efflux and

simulated rhizosphere respiration in a beech forest. Tree Physiology 21:145–152. doi: 10.1093/treephys/21.2-3.145 Ruehr NK, Buchmann N (2010) Soil respiration fluxes in a temperate mixed forest: seasonality and temperature sensitivities differ among microbial and root-rhizosphere respiration. Tree Physiol 30:165–176.

(3) The third point deals with estimate the number of samples required for a robust estimate of the Rs. This has also been done plenty of time so there are two options: use it as a description of the site in the materials and methods section or do not only compare with other estimates but discuss more the reason why the number of samples required is higher in this study than in many others, thus why spatial variability is higher. Four lines is not enough. The discussion now is poor.

In conclusion, while the manuscript is based on an interesting data set obtained with valid methods, the discussion is not strong enough to reach substantial conclusions. A little more time would have been needed, maybe. One may expect the last sentence of the abstract to be the core of the discussion. The state of the art in the introduction should also be reinforced by looking more in details in the huge relevant literature. The argument that no study has examined the influences of trees on spatial variation of Rs in the Chesapeake Bay watershed can be used for millions of watersheds in the world. This sentence should be removed.

Specific comments:

Line 21 (and 47): remove "in time" there is no evidence that a better knowledge of spatial variation will improve scaling soil respiration "in time"

Line 35: need a clear definition of what is stand. In the description, there are 3 sites and 3 plots within site, but no stand.

Lines 64-66: This sentence is very speculative, probably wrong and not needed. At similar age, tropical forests are at least as productive as temperate forest, and evergreen forests in a given climate are at least as productive as deciduous forests on an

annual basis.

Line 107: linear or exponential regression. . . which one is reported?

Line 122: why not try to consider the size of the tree when increasing the radial distance. I mean include only big trees when far from the collar and all trees when close to the collar. Why not testing all distances from 1 to 15 m and use the one that give the best correlation with soil respiration?

Lines 125-139: provide the model. According to Table 3, soil moisture has a specific equation. Which non-significant terms were eliminated using the forward-and-back stepwise algorithm. This is not so clear that terms have been removed when looking at Table 3

Lines 140-145: it will be better to add the type of season as an additional factor in your model rather than running the model on a split dataset. Same comment for the dryness splitting.

Line 173: 40% is not almost half. But maybe it is 49%, not 40 (table 3)

Lines 185-187: not need to recall hypothesis in the result section.

Lines 195-201: the analysis will be greatly improved by considering not only the distance but also the size of the neighbour trees that interact with the distance (bigger trees have influence on longer distance)

Lines 293: the data does not support the idea that the high spatial variation is related to stand structure. First, only BA is considered for characterising stand structure, which probably is not enough. And second, this BA does not explain so much the variation.

Table 1: add the altitude of the three sites so that the risk of submersion can be evaluated by the readers

Table 2: do not use +/- for SD, this is statistically incorrect (OK with SE or CI only). Instead use parentheses.

Table 3: Improve the presentation (less digit, SE in the same column than value with +/-). Check df values, there is a problem. And show, in addition to the global model, the all the sub models you use (dormant versus growing). And the three dryness thirds as well. But see previous comments on the model.

Figure 1 is not very informative (not useful)

Figure 2: why mean flux? It is called individual observation in Fig 4 which seems better

Figure 3: hard to see what happens at short distance, especially at 5 m that is the selected distance. Can a log scale improve readability?

Figure 4: check x-axis labelled

---

## Referee Comment (RC2) · Anonymous Referee #2 · 26 Jul 2019

This paper addresses a current knowledge gap with forest soil respiration research: how important is the presence of vegetation for helping to explain some of the variability in soil respiration over space? We often treat forests as homogeneous when designing field studies. However, there is an accumulation of research that suggests that the spatial arrangement, size and density of trees can affect soil respiration measured in a particular spot. It's important to be able to characterize this effect for many reasons, which the authors point out - designing the spatial arrangement of measurements, interpreting relationships of soil respiration with environmental variables and seasonality, to name a few. I appreciate the authors' study design, especially their decision to sample sub-monthly and not just focusing on the growing season but also reporting results

from the dormant season. The paper has some weaknesses that dilute the impact of the study, I believe, that can be improved. There are also some omissions that should be included, and some of the statistical approach should be re-considered. The paper is generally well-written but (like most papers) could use some further clarification in places.

I have concerns with the title of the paper that affects some of the text in the paper and the way the problem is framed and studied. 'Tree proximity' implies that the research is focused on understanding how the degree of closeness of trees to soil respiration measurement influences respiration. This is not what the study is doing. Rather, I think a more accurate title would be something like "localized basal area affects soil respiration dynamics in a coastal temperate deciduous forest". This is because the only variable included in the models that involves trees is basal area within a 5 m radius, and the focus of the statistical modeling was on determining if localized basal area had an effect in addition to temperature and moisture. Based on the title, I was expecting a different kind of analysis, such as kriging or a spatial regression. 'Tree proximity' could be interpreted to mean different kinds of things. 'Localized basal area' is more specific to the actual variable that was examined.

Parts of the discussion and conclusion involve making assumptions about autotrophic and heterotrophic respiration based on their findings. It is tempting to make these statements (I've been there before), but you have to be careful here. Trees do not just influence autotrophic respiration - they provide fresh substrate for heterotrophic respiration as well. I think it is okay to include some speculation of how tree presence/absence might influence respiration rates, but try to avoid the assumption that trees only affect the autotrophic side of things.

The statistical methods used to determine whether variables were stronger or weaker and to compare dormant season model fit to growing season model fit should be re-examined. Differences in $R^2$ and AIC between models that use different input data do not necessarily indicate that the fit is better or worse. You could look into using an ef-
fect size analysis or examining relative importance of regression parameters (package relaimpo in R).

Overall, this is an interesting study that investigates the influence of localized basal area on soil respiration - with some improvements, this should be an impactful contribution to the literature. Keep up the good work.

Line by line edits: 23: I would remove all mentions of 'proximity' (since you aren't measuring how close each tree is to the collars) and replace with something more descriptive. 'Presence' would work here. 24: Again, I would replace 'tree proximity' with 'localized basal area'. 26: Needs to be more specific - within a 5m 'radius' 46: This statement is misleading - large whole-tree chambers, for example, are in effect measuring soil respiration at scales larger than 1 m. Re-write this sentence to better say what you mean. 74: 'Higher' would probably be better than 'stronger' 90-101: A description of the soils present at the sites should be included. 99: Need more detail here - was the separation distance randomly selected, the direction from plot center, or...? 132: I don't think you need the second 'h' in homoscedasticity. I could be wrong though. 145: It would help the reader if you explicitly stated what you were looking for in your models (even though it should be obvious). For example: 'a higher parameter for BA5 in the dormant season model would support the hypothesis that...' 227: This is confusing - how would you have a temperature effect after controlling for temperature effects? 227: I disagree - you cannot attribute an effect of trees solely to autotrophic respiration. Tree presence is also correlated with substrate for microbes. 246: This is another assumption that is not well-supported - that trees do not have shallow roots that contribute to respiration. 248: I suggest searching the literature for more references on soil moisture restrictions on respiration from deciduous forests. Here are a couple to get you started:

Contosta, A.R., Burakowski, E.A., Varner, R.K., Frey, S.D., 2016. Winter soil respiration in a humid temperate forest: the roles of moisture, temperature, and snowpack. J. Geophys. Res. Biogeosci. 2016JG003450. https://doi.org/10.1002/2016JG003450

[Figure]

Jiang, H., Deng, Q., Zhou, G., Hui, D., Zhang, D., Liu, S., Chu, G., Li, J., 2013. Responses of soil respiration and its temperature/moisture sensitivity to precipitation in three subtropical forests in southern China. Biogeosciences 10, 3963–3982.

288: Autotrophic and heterotrophic respiration were not partitioned in this study - please revise this sentence to better communicate the overall finding of your work. 479: Column header abbreviations should be defined. 'DF' should be clarified as 'denominator degrees of freedom'.
* * *

---

## Referee Comment (RC3) · Anonymous Referee #3 · 15 Aug 2019

Pennington et al report on a well designed study of the effect of nearby trees on soil respiration, which finds that nearby trees increase the temperature sensitivity (but not the rate) of soil respiration except during dry periods and during the dormant season. The topic is important and appropriate to the journal, and the article is very well written. It was a pleasure to read.

My general criticism is that the article does not discuss what for me is the 'elephant in the results'. The authors take the effect of BA5 on the T sensitivity of Rs to mean that Ra is more sensitive than Rh to T. The logic is that when there is more nearby basal area (and hence, by assumption, root biomass), Ra is a greater component of

Rs. However, BA5 was not found to be a significant driver of the spatial variability of Rs. It seems to me that the only way to reconcile those two ideas is to suppose that as root biomass increases, Ra increases but Rh decreases by the same amount in order to keep Rs the same, which as far as I know is not something that is believed to happen. If anything, the literature suggests that root exudates fuel soil respiration, rather than competing with it. I therefore think the discussion needs to acknowledge this paradox and tackle the question of how roots could plausibly impact the T sensitivity of Rs without impacting the magnitude of Rs. Could it be some kind of statistical artifact? The article doesn't necessarily have to have the answer, but it should at least lay out the key questions and suggest what kind of further work might be able to answer them.

Specific

lines 34-5: "We estimate that four RS observations were required to be within 50% of the stand-level mean, and 311 to be within 5%, at 90% confidence." After reading the article, the meaning of this sentence became clear, but when it is first encountered in the abstract, its grammatical ambiguity causes it to sounds like nonsense (who required your observations to be close to the mean, and what does it mean to be within 50% of a mean?). I would rephrase it to something like: "Due to that variability, we determine that four RS observations would be required in order to estimate the stand level mean to within 50%, and 311 would be required in order to estimate it to within 5%, at 90% confidence."

line 50: What is "leaf habit"? And does ecosystem-scale productivity really affect the sub-ecosystem-scale spatial variability, or do you mean to say something different?

lines 133-4: It seems to me that taking the log shouldn't turn heteroschedasticity into homoschedasticity: if the variability in Rs varies with Rs, then the variability in log(Rs) will also vary with log(Rs), no?

line 213: I think "temperature" should be "temperate".

[Figure]

Figure 4: This figure's axes are swapped. Right now it is basal area vs residual (not residual vs basal area as the caption says). The residual is the independent variable that should be on the x axis. More importantly, it looks like the regression lines were calculated with this reversal of dependent and independent variables as well, so that all variability is attributed to the basal area measurement (which is actually quite precise I'm sure) rather than to the respiration (which is actually quite noisy). Also, there is a strange character in the x axis label.

---

## Author Comment (AC3) · 12 Sep 2019

1   "Pennington et al report on a well designed study of the effect of nearby trees on soil respiration, which finds that nearby trees increase the temperature sensitivity (but not the rate) of soil respiration except during dry periods and during the dormant season.  The topic is important and appropriate to the journal, and the article is very well written. It was a pleasure to read."

*Thank you for the kind words.*

2   **"My general criticism is that the article does not discuss what for me is the 'elephant in the results'. The authors take the effect of BA5 on the T sensitivity of Rs to mean that Ra is more sensitive than Rh to T. The logic is that when there is more nearby basal area (and hence, by assumption, root biomass), Ra is a greater component Rs. However, BA5 was not found to be a significant driver of the spatial variability of Rs. It seems to me that the only way to reconcile those two ideas is to suppose that as root biomass increases, Ra increases but Rh decreases by the same amount in order to keep Rs the same, which as far as I know is not something that is believed to happen. If anything, the literature suggests that root exudates fuel soil respiration, rather than competing with it. I therefore think the discussion needs to acknowledge this paradox and tackle the question of how roots could plausibly impact the T sensitivity of Rs without impacting the magnitude of Rs. Could it be some kind of statistical artifact? The article doesn't necessarily have to have the answer, but it should at least lay out the key questions and suggest what kind of further work might be able to answer them."**

*Thank you for pointing this out. This is not an artifact, but it is a consequence of the model we used, which was not well communicated. Localized basal area entered the model as a fixed effect (i.e., testing whether it raised Rs by itself; this was not significant) and in an interaction with temperature (testing whether it changed temperature sensitivity; this was significant). Because, over the course of the day/month/year, temperature varies significantly in these forests, the result is that the changed temperature sensitivity results in a higher cumulative Rs flux for collars in high-BA locations. This can be seen by plotting the raw respiration data and fitting a loess curve, separating the data into low- and high-BA colors (see Fig. 1).(Conversely, one could imagine a situation where these lines were perfectly parallel throughout the year. In this case the BA effect would be significant, but there would be no difference in the temperature sensitivity.)*

**3   OVERALL RESPONSE:**

*Thanks for the thoughtful comments. We will clarify the point made in 2 in the text, and think that including a version of Fig. 1 in the revised manuscript will help readers understand the practical consequences of the statistical model.*

**Fig. 1.** Flux in ($\mu$mol CO2 mˆ-2 sˆ-1) is shown by day of the year. Collars that "see" high basal area are in red, and low in blue (cut off by the median BA value) with a loess fit.

---

## Author Response (AR1)

**Referee Comments and Responses**

**Anonymous Referee #1**

The manuscript entitled "Tree proximity affects soil respiration dynamics in a coastal temperate deciduous forest" is in fact addressing three different questions: (1) tree proximity and soil respiration, (2) temperature sensitivity, and (3) required sampling effort. Only the first one is clearly reflected in the title.

> *This is a great point, also pointed out by Reviewer #2. We will change the title to something that reflects the overall Rs variability in the context of localized basal area affect/vegetation. For example, Reviewer #2 suggested "localized basal area affects soil respiration dynamics in a coastal temperate deciduous forest", which we agree would work better than the current title.*

These three questions are relevant and within the scope of BG, but they are not novel and there is no novel concept, idea or tool that emerged for this study. This is an additional set of data (a case study). (1) The approach of linearly connecting the basal area of trees to a fixed distance (5 m) and ground respiration is simplistic. The distance at which an individual tree influenced soil respiration is probably dependent on the size of this tree. In other words, biggest trees are expected to have a stronger influence than smaller trees.

> *While aspects of the relationship between Rs and basal area have been previously studied, the issue is hardly closed; there is little consensus on the strength and spatial patterns of this effect, for example. We believe that the novelty of this study lies in its examination of how basal area affects the spatial variability of Rs in different phenological seasons and soil moisture conditions. We will explain this more clearly in the revised manuscript.*

There are several (many papers) relating addressing the effect of tree size and proximity on soil respiration that are not cited in this manuscript. Among them: Fang C, Moncrieff JB, Gholz HL, Clark KL (1998) Soil CO2 efflux and its spatial variation in a Florida slash pine plantation. Plant Soil 205:135–146. doi:10.1023/A:1004304309827 Metcalfe DB, Meir P, Aragão LEOC, Malhi Y, da Costa ACL, Braga A, Gonçalves PHL, de Athaydes J, de Almeida SS, Williams M (2007) Factors controlling spatio-temporal variation in carbon dioxide efflux from surface litter, roots, and soil organic matter at four rain forest sites in the eastern Amazon: PARTITIONING AMAZON SOIL RESPIRATION. J Geophys Res 112. doi:10.1029/2007JG000443 Katayama A, Kume T, Komatsu H, Ohashi M, Naka- gawa M, Yamashita M, Otsuki K, Suzuki M, Kumagai T (2009) Effect of forest structure on the spatial variation in soil respiration in a Bornean tropical rainforest. Agric For Meteorol 149:1666–1673. doi :10.1016/j.agrformet.2009.05.007 Bréchet L, Ponton S, Alméras T, Bonal D, Epron D (2011) Does spatial distribution of tree size account for spatial variation in soil respiration in a tropical forest? Plant and Soil 347:293–303. doi: 10.1007/s11104-011-0848-1 Schwendenmann L, Macinnis-Ng C (2016) Soil CO2 efflux in an

old-growth southern conifer forest (Agathis australis) – magnitude, com- ponents and controls. SOIL 2:403–419. doi: 10.5194/soil-2-403-2016 Reading these papers (but the list is not limitative) would have given way to analyze more finely the results, especially the last two.

*We appreciate these suggestions, and agree that additional citation of the literature would strengthen the discussion.*

(2) The observation that autotrophic respiration is more sensitive to temperature than heterotrophic respiration is also confirmative of many studies. Note that the paper Aguilos et al 2011 that is cited when discussing this point has not been accepted for publication in Biogeoscience, so the citation is wrong. Note that the citation Wei et al is incorrect: should be Wei et al (doi: 10.1016/j.soilbio.2010.04.013).

*Thank you for catching these mistakes, which will be fixed.*

The discussion of this fact is rather poor and miss one of the most important drivers of the apparent temperature sensitivity of RA: phenology. This may be important in the present study since soil respiration was measured over a full year and species are deciduous. Among many other sources, this has been discussed in: Epron D, Le Dantec V, Dufrêne E, Granier A (2001) Seasonal dynamics of soil carbon dioxide efflux and simulated rhizosphere respiration in a beech forest. Tree Physiology 21:145–152. doi: 10.1093/treephys/21.2-3.145 Ruehr NK, Buchmann N (2010) Soil respiration fluxes in a temperate mixed forest: seasonality and temperature sensitivities differ among microbial and root-rhizosphere respiration. Tree Physiol 30:165–176.

*We agree, a discussion of phenological influence will help to explain the high spatial variability at this site and temperature sensitivity found.*

(3) The third point deals with estimate the number of samples required for a robust estimate of the Rs. This has also been done plenty of time so there are two options: use it as a description of the site in the materials and methods section or do not only compare with other estimates but discuss more the reason why the number of samples required is higher in this study than in many others, thus why spatial variability is higher. Four lines is not enough. The discussion now is poor.

*Thank you for pointing this out. While we agree the sample requirement section should be better addressed, we propose to leave this in the discussion for three reasons. The spatial variability could be a product of 1) the topographic variability at the study site led some collars to be better drained than others, 2) the record rainfall year in 2018, and 3) species diversity. All which could contribute to the high variability and thus high number of samples required.  We will expand beyond the original four lines of text to better cover sampling challenges and solutions.*

In conclusion, while the manuscript is based on an interesting data set obtained with valid methods, the discussion is not strong enough to reach substantial conclusions. A little more time would have been needed, maybe. One may expect the last sentence of the abstract to be the core of the discussion. The state of the art in the introduction should also be reinforced by looking more in detail in the huge relevant literature. The argument that no study has examined the influences of trees on spatial variation of Rs in the Chesapeake Bay watershed can be used for millions of watersheds in the world. This sentence should be removed.

*We agree, and will remove this sentence.*

**OVERALL RESPONSE TO R1:**

*Thank you for your review. Overall, we agree that (1) the title should be reconsidered to include the entire scope of the study (also pointed out by Review 2), (2) better acknowledgement of current literature will give the study motivation more context and (3) a more in-depth consideration of phenology will strengthen the discussion. However, while the relationship between Rs and basal area has been previously cited, we believe the novelty of this study lies in the examination of how vegetation influences the spatial variability in forest ecosystems.*

Specific comments:
Line 21 (and 47): remove "in time" there is no evidence that a better knowledge of spatial variation will improve scaling soil respiration "in time"

*We agree and have removed "in time" where necessary, lines 20 and 48.*

Line 35: need a clear definition of what is stand. In the description, there are 3 sites and 3 plots within site, but no stand.

*Here a "stand" can be considered equivalent to a "site". We have clarified this point in the text (line 34).*

Lines 64-66: This sentence is very speculative, probably wrong and not needed. At similar age, tropical forests are at least as productive as temperate forest, and evergreen forests in a given climate are at least as productive as deciduous forests on an annual basis.

*We have revised this sentence in lines 65-67 to note merely that these are productive, mid-latitude deciduous forests and thus autotrophic effects on Rs might be particularly strong.*

Line 107: linear or exponential regression. . . which one is reported?

*Good catch; yes, the IRGA reports both. For this analysis, we used the exponential regression values. The manuscript has been updated in line 107 to reflect this.*

Line 122: why not try to consider the size of the tree when increasing the radial distance. I mean include only big trees when far from the collar and all trees when close to the collar.

*This would have been an interesting approach, balancing sampling efficiency with including very local effects of small trees, and similar in spirit to that employed by Bréchet et al. (2011). We appreciate the suggestion and will consider it for our next study.*

Why not testing all distances from 1 to 15 m and use the one that give the best correlation with soil respiration?

*This would definitely have been possible. We preferred to start with a biologically-driven hypothesis of 5 m (effect radius) and test that, before proceeding to test all possible differences (i.e. in Figure 5). Both approaches have their strengths but in this case we think a hypothesis-driven framework is the right choice, providing a clear test and straightforward interpretation.*

Lines 125-139: provide the model. According to Table 3, soil moisture has a specific equation. Which non-significant terms were eliminated using the forward-and-back stepwise algorithm. This is not so clear that terms have been removed when looking at Table 3

*We now provide the full model specification in the methods, line 134. This provides a useful reference against the model results given in Table 3.*

Lines 140-145: it will be better to add the type of season as an additional factor in your model rather than running the model on a split dataset. Same comment for the dryness splitting.

*This would have been a good technique–thank you. Here, however, following Reviewer 2's advice, we have elected to keep the separate model approach paired with a robust measurement of variable importance using R's 'relaimpo' package.*

Line 173: 40% is not almost half. But maybe it is 49%, not 40 (table 3)

*Thank you for the comment, the model predicted 37% of the variability, so we changed the phrase "almost half" to reflect this in line 173.*

Lines 185-187: not need to recall hypothesis in the result section.

*We have removed the first sentence of this paragraph for clarity and concision (lines 185-190).*

Lines 195-201: the analysis will be greatly improved by considering not only the distance but also the size of the neighbour trees that interact with the distance (bigger trees have influence on longer distance)

*Please see our response to the line 122 comment above.*

Lines 293: the data does not support the idea that the high spatial variation is related to stand structure. First, only BA is considered for characterising stand structure, which probably is not enough. And second, this BA does not explain so much the variation.

*We respectfully disagree. We are not saying that BA controls \*all\* spatial Rs variation, just that it is indisputably linked to it.*

Table 1: add the altitude of the three sites so that the risk of submersion can be evaluated by the readers

*Thank you for the comment, Table 1 now has a column for altitude ranges at our three sites, see line 502.*

Table 2: do not use +/- for SD, this is statistically incorrect (OK with SE or CI only). Instead use parentheses.

*We have replaced +/- with parentheses, line 510.*

Table 3: Improve the presentation (less digit, SE in the same column than value with +/-). Check df values, there is a problem. And show, in addition to the global model, the all the sub models you use (dormant versus growing). And the three dryness thirds as well. But see previous comments on the model.

*Thank you for your comment. We have removed a digit and put SE in the same column with value as you suggested (line 516).*

Figure 1 is not very informative (not useful)

*In explaining our study design at numerous meetings and conferences, we have found that people frequently get confused about whether our measurements were conducted with respect to individual trees or individual Rs measurement points. For this reason, and because studies of tree proximity and RS more typically measure from trees or in a transect (Tang, J. and Baldocchi, 2005), we prefer to keep Figure 1.*

Figure 2: why mean flux? It is called individual observation in Fig 4 which seems better

*At each measurement point, the IRGA took two consecutive measurements. By "mean flux" we meant that the two consecutive measurements were averaged. This is stated in the methods section (lines 106-109) and has been clarified in the caption of Figure 2 (line 540).*

Figure 3: hard to see what happens at short distance, especially at 5 m that is the selected distance. Can a log scale improve readability?

*A log scale did not improve readability enough, however we have decreased the line size and added a subplot of 0-5 (line 545).*

Figure 4: check x-axis labelled

*Thank you. We have corrected this in line 552.*

**Anonymous Referee #2**

This paper addresses a current knowledge gap with forest soil respiration research: how important is the presence of vegetation for helping to explain some of the variability in soil respiration over space? We often treat forests as homogeneous when designing field studies. However, there is an accumulation of research that suggests that the spatial arrangement, size and density of trees can affect soil respiration measured in a particular spot. It's important to be able to characterize this effect for many reasons, which the authors point out - designing the spatial arrangement of measurements, interpreting relationships of soil respiration with environmental variables and seasonality, to name a few. I appreciate the authors' study design, especially their decision to sample sub-monthly and not just focusing on the growing season but also reporting results from the dormant season. The paper has some weaknesses that dilute the impact of the study, I believe, that can be improved. There are also some omissions that should be included, and some of the statistical approach should be re-considered. The paper is generally well-written but (like most papers) could use some further clarification in places.

*Thanks for the thoughtful comments and assessment.*

I have concerns with the title of the paper that affects some of the text in the paper and the way the problem is framed and studied. 'Tree proximity' implies that the research is focused on understanding how the degree of closeness of trees to soil respiration measurement influences respiration. This is not what the study is doing. Rather, I think a more accurate title would be something like "localized basal area affects soil respiration dynamics in a coastal temperature deciduous forest". This is because the only variable included in the models that involves trees is basal area within a 5 m radius, and the focus of the statistical modeling was on determining if localized basal area had an effect in addition to temperature and moisture. Based on the title, I was expecting a different kind of analysis, such as kriging or a spatial regression. 'Tree

proximity' could be interpreted to mean different kinds of things. 'Localized basal area' is more specific to the actual variable that was examined.

> *This is a great point and also noted by Reviewer 1. Your title suggestion is a good one; we will change the title to something that reflects the overall Rs variability in the context of localized basal area affect/vegetation.*

Parts of the discussion and conclusion involve making assumptions about autotrophic and heterotrophic respiration based on their findings. It is tempting to make these statements (I've been there before), but you have to be careful here. Trees do not just influence autotrophic respiration - they provide fresh substrate for heterotrophic respiration as well. I think it is okay to include some speculation of how tree presence/absence might influence respiration rates, but try to avoid the assumption that trees only affect the autotrophic side of things.

> *This is a fair point - upon revision, we will clarify that these are broad assumptions but may not reflect the complex real-world links between Ra and Rh. We also believe that (as pointed out by Review 1) a further discussion of phenology will allow us to acknowledge other processes that may influence Rs.*

The statistical methods used to determine whether variables were stronger or weaker and to compare dormant season model fit to growing season model fit should be re-examined. Differences in R2 and AIC between models that use different input data do not necessarily indicate that the fit is better or worse. You could look into using an effect size analysis or examining relative importance of regression parameters (package relaimpo in R).
Overall, this is an interesting study that investigates the influence of localized basal area on soil respiration - with some improvements, this should be an impactful contribution to the literature. Keep up the good work.

> *This is absolutely correct; our initial approach here is vulnerable, as you note, to differences in dataset size and other factors. We appreciate the introduction to the 'relaimpo' package, and will use it, or an equivalent approach, to robustly examine the relative importance of model terms in our linear regression analyses.*

**OVERALL RESPONSE TO R2:**

> *Thank you for the critique. To best address your suggestions, we will (1) create a new title that better reflects the purpose of the study, (2) clarify assumptions being made, especially in regards to Rh and Ra drivers and the links between them, and (3) change our statistical analysis to more robustly compare models of differing sample sizes, especially between our growing and dormant season models.*

Line by line edits:

23: I would remove all mentions of 'proximity' (since you aren't measuring how close each tree is to the collars) and replace with something more descriptive. 'Presence' would work here.

*Thank you and we agree, this wording does not reflect the entire scope of the study. We have replaced the mentions of 'proximity' in lines 21 and 23.*

24: Again, I would replace 'tree proximity' with 'localized basal area'.

*Please see our response to the line 23 comment above.*

26: Needs to be more specific - within a 5m 'radius'

*This is revised in line 25.*

46: This statement is misleading - large whole-tree chambers, for example, are in effect measuring soil respiration at scales larger than 1 m. Re-write this sentence to better say what you mean.

*We have tweaked the sentence (lines 46-48); but feel obliged to note that 'whole tree chambers' measure plant respiration \*in addition to\* soil respiration, and it's not straightforward to separate the two. (Similarly, by this logic, eddy covariance towers 'measure' Rs too.) We believe it is accurate, however, to say that Rs can't be directly measured, all by itself, at scales larger than ~1 $m^2$.*

74: 'Higher' would probably be better than 'stronger'

*This is a good point and has been revised on line 75.*

90-101: A description of the soils present at the sites should be included.

*Thank you for the comment. We included the dominant soil types at each site in Table 1 (line 502).*

99: Need more detail here - was the separation distance randomly selected, the direction from plot center, or. . .?

*The collars were randomly placed within each plot (line 99).*

132: I don't think you need the second 'h' in homoscedasticity. I could be wrong though.

*Yes, you are correct. This has been revised in lines 132 and 140.*

145: It would help the reader if you explicitly stated what you were looking for in your models (even though it should be obvious). For example: 'a higher parameter for BA5 in the dormant season model would support the hypothesis that. . .'

> *We have expanded the sentence in lines 145-146 to state that we are looking for significance in the BA5 parameter of our model. We hope this makes our methods more explicit.*

227: This is confusing - how would you have a temperature effect after controlling for temperature effects?

> *Thank you for the comment. We have rephrased this finding for clarity in lines 229-231 and 298-299.*

227: I disagree - you cannot attribute an effect of trees solely to autotrophic respiration. Tree presence is also correlated with substrate for microbes.

> *We agree with you. Because we did not directly measure Ra and Rh contributions, we cannot attribute the BA5 effect solely to Ra. We have changed the wording to reflect this statement as a hypothesis as well as added that trees (BA) also contributes to Rh. See lines 231-236.*

246: This is another assumption that is not well-supported - that trees do not have shallow roots that contribute to respiration.

> *We don't suggest that trees don't have shallow respiring roots, but rather that they probably have access to deeper water sources. We respectfully suggest to leave the wording of our hypothesis as is.*

248: I suggest searching the literature for more references on soil moisture restrictions on respiration from deciduous forests. Here are a couple to get you started: Contosta, A.R., Burakowski, E.A., Varner, R.K., Frey, S.D., 2016. Winter soil respiration in a humid temperate forest: the roles of moisture, temperature, and snowpack. J. Geophys. Res. Biogeosci. 2016JG003450. https://doi.org/10.1002/2016JG003450 Jiang, H., Deng, Q., Zhou, G., Hui, D., Zhang, D., Liu, S., Chu, G., Li, J., 2013. Responses of soil respiration and its temperature/moisture sensitivity to precipitation in three subtropical forests in southern China. Biogeosciences 10, 3963–3982.

> *Thank you for these suggestions. We have added more referenced on soil restrictions, specifically for deciduous forests, to reinforce our results. (see lines 219-224 and lines 247-252).*

288: Autotrophic and heterotrophic respiration were not partitioned in this study - please revise this sentence to better communicate the overall finding of your work.

*We have removed the first part of this sentence since Rh and Ra were not partitioned (lines 298-299).*

479: Column header abbreviations should be defined. 'DF' should be clarified as 'denominator degrees of freedom'.

> *We have defined 'DF' in the revised manuscript for clarity, which can be found in line 516.*

**Anonymous Referee #3**

Pennington et al report on a well designed study of the effect of nearby trees on soil respiration, which finds that nearby trees increase the temperature sensitivity (but not the rate) of soil respiration except during dry periods and during the dormant season. The topic is important and appropriate to the journal, and the article is very well written. It was a pleasure to read.

> *Thanks for the kind words, and thoughtful comments.*

My general criticism is that the article does not discuss what for me is the 'elephant in the results'. The authors take the effect of BA5 on the T sensitivity of Rs to mean that Ra is more sensitive than Rh to T. The logic is that when there is more nearby basal area (and hence, by assumption, root biomass), Ra is a greater component Rs. However, BA5 was not found to be a significant driver of the spatial variability of Rs. It seems to me that the only way to reconcile those two ideas is to suppose that as root biomass increases, Ra increases but Rh decreases by the same amount in order to keep Rs the same, which as far as I know is not something that is believed to happen. If anything, the literature suggests that root exudates fuel soil respiration, rather than competing with it. I therefore think the discussion needs to acknowledge this paradox and tackle the question of how roots could plausibly impact the T sensitivity of Rs without impacting the magnitude of Rs. Could it be some kind of statistical artifact? The article doesn't necessarily have to have the answer, but it should at least lay out the key questions and suggest what kind of further work might be able to answer them.

**OVERALL RESPONSE TO R3:**

> *Thank you for pointing this out. This isn't an "artifact", but it is a consequence of the model we used, that was not well communicated. Localized basal area entered the model as a fixed effect (i.e., testing whether it raised RS by itself; this was not significant) and in an interaction with temperature (testing whether it changed temperature sensitivity; this was significant). Because, over the course of the day/month/year, temperature varies significantly in these forests, the result is that the changed temperature sensitivity results in a higher cumulative RS flux for collars in high-*

*BA locations. This can be seen by plotting the raw respiration data and fitting a **spline** curve, separating the data into low- and high-BA colors:*

[Figure]

*(Conversely, one could imagine a situation where these lines were perfectly parallel throughout the year. In this case the BA effect would be significant, but there would be no difference in the temperature sensitivity.)*

*We will clarify this point in the text, and think including a version of this figure in the revised manuscript will help readers understand the practical consequences of the statistical model.*

Specific lines

34-5: "We estimate that four RS observations were required to be within 50% of the stand-level mean, and 311 to be within 5%, at 90% confidence." After reading the article, the meaning of this sentence became clear, but when it is first encountered in the abstract, its grammatical ambiguity causes it to sounds like nonsense (who required your observations to be close to the mean, and what does it mean to be within 50% of a mean?). I would rephrase it to something like: "Due to that variability, we determine that four RS observations would be required in order to estimate the stand level mean to within 50%, and 311 would be required in order to estimate it to within 5%, at 90% confidence."

*We agree and have revised the sentence for clarity based on your suggestion, thank you. Please see lines 33-35.*

line 50: What is "leaf habit"? And does ecosystem-scale productivity really affect the sub-ecosystem-scale spatial variability, or do you mean to say something different?

*"Leaf habit" denotes whether a tree or ecosystem is deciduous or evergreen. We have clarified this in the text (lines 50-53).*

lines 133-4: It seems to me that taking the log shouldn't turn heteroschedasticity into homoschedasticity: if the variability in Rs varies with Rs, then the variability in log(Rs) will also vary with log(Rs), no?

*Thank you for the comment, taking the log of the dependent variable is a standard approach to mitigate heteroscedasticity; see e.g.* [https://en.wikipedia.org/wiki/Heteroscedasticity#Fixes](https://en.wikipedia.org/wiki/Heteroscedasticity#Fixes)*. However, you are correct that the variability growth is still present after transformation, but it's so slow that it's no longer causing problems for least squares.*

line 213: I think "temperature" should be "temperate".

*Good catch! We have corrected this in line 211.*

Figure 4: This figure's axes are swapped. Right now it is basal area vs residual (not residual vs basal area as the caption says). The residual is the independent variable that should be on the x axis. More importantly, it looks like the regression lines were calculated with this reversal of dependent and independent variables as well, so that all variability is attributed to the basal area measurement (which is actually quite precise I'm sure) rather than to the respiration (which is actually quite noisy). Also, there is a strange character in the x axis label.

*We have fixed axes and label errors in Figure 4, see line 552. The regression lines were calculated as Rs ~ BA.*

[revised manuscript text omitted]

--- END OF MANUSCRIPT ---

---

## Author Response (AR2)

For final publication, the manuscript should be:
      **accepted as is**

Suggestions and responses:

The manuscript has been nicely improved after the revision and most of the suggestions I did have been taken into consideration.
Reading in your answer that you are considering the possibility of improving the model by including the size of the tree in a future study reinforces my belief that it could have been done in this one as well, and that it would have been great and would have given more value to your study.
I do not understand in your answer why it makes sense to write that a radius of 5 m is a hypothesis dictated by biology. But fortunately, this does not appear in the revised manuscript and the new figure 5 is more convincing than the previous one

*Thank you for your comments and we appreciate the feedback.*

**Anonymous Referee #3**

For final publication, the manuscript should be:
      **accepted subject to technical corrections**

Suggestions and responses:

Pennington et al. have provided a nice revision of their original manuscript, which successfully addresses my concerns except for these relatively minor issues:

- I was sorry to see that some version of the figure from the authors' "OVERALL RESPONSE TO R3" did not make it into the revised draft. I found it helpful for understanding how BA5 might impact Rs via T without having a statistically significant effect on Rs directly. However, if the authors and editor do not feel the need for such a figure, then I do not object.

*We believe that while the figure we provided in the response provided a valuable explanation for the Rs:T:BA significance result, it did not add enough information to the whole "story" of localized basal area effect but think it will be a good addition to the supplemental information section.*

- lines 57 ff.: The phrasing of this paragraph (particularly the first and last sentences) portrays the influence of vegetation distribution on Rs as a hypothesis to be tested, whereas the citations show that root respiration has already been shown to contribute substantially to Rs, and Rs has already been shown to be higher near tree stems — i.e. the spatial distribution of vegetation has already been shown to affect the spatial distribution of Rs. Additionally, the logic of the first sentence is flawed: the fact that "[a]t large scales, Rs differs between vegetation types and biomes" does not even imply that vegetation type affects Rs (it could be that soil type or other factors affect both

vegetation type and Rs), let alone "that the spatial distribution of vegetation might strongly affect Rs via plant root respiration" (which involves logical leaps to spatial distribution and root respiration). This is the paragraph that needs to clearly lay out what's already known about the influence of vegetation on Rs and state the knowledge gap that this study is going to fill, which is not really about the spatial distribution of Rs, but rather about how vegetation affects the response of Rs to environmental drivers (especially temperature).

> *This is a good point, and we have reworked this paragraph for logical consistency, and to emphasize that while it's clearly established that Rs varies spatially due to vegetation, much less is understood about the implications of this for the overall Rs sensitivity to environmental conditions.*

- lines 111-114: Soil moisture and temperature measurements don't belong in a section called "Soil respiration measurements".

> *We have changed the title of this subsection to "Soil respiration and ancillary measurements".*

- line 116: Strictly speaking, these are not "tree proximity measurements", but rather local basal area measurements.

> *We have changed the title of this subsection to "Local basal area measurements" to match the wording in the title and study description.*

- line 132: I acknowledge the author's response regarding heteroscedasticity and the logarithm, but I think "ensure homoscedasticity" is too strong language here. I would say that taking the logarithm dampens heteroscedasticity.

> *We have changed the wording here.*

- line 134: I think this "equation" requires a bit of justification/derivation. E.g. why SM and SM^2? Does this notation mean that log(Rs) is taken to be linearly related to T5*BA5 and to T20*BA5 but quadratically related to SM? Why?

> *We have clarified this point in the methods. Basically, over the course of a year, we expect that Rs might be limited by both too little SM and too much SM–i.e., it would respond both positively and negatively to SM changes depending on the degree of soil anoxia. For this reason a quadratic SM term is included in the model. In this forested temperate ecosystem we don't expect to ever see a decline of Rs with temperature increase, however.*

- line 297 ff.: The conclusion is awkward. Mention of Rs is left out of the first sentence, about the key finding. In the second sentence, "these sites" is ambiguous as to whether you're referring to all your study sites or just the sites with high BA5. In the third sentence, the fact that "soil respiration at this site [is] highly dynamic and variable" is a direct observation, not something suggested by the findings.

*We have revised the wording in this paragraph to better highlight Rs findings and to clarify that Rs variability (a direct observation) may contribute to the localized basal area influence we found and high sample requirements at our study sites (suggested by findings).*

- Fig. 3: The inset adds no information and should be removed. The color scale could be put in its place instead, reduce the overall space required by the figure.

*Thank you for the feedback, we've replaced the inset graph with the color scale.*

-------- END OF RESPONSES -------

Localized basal area affects soil respiration temperature sensitivity in a coastal deciduous forest

Biogeosciences

Stephanie C. Pennington*[1], Nate G. McDowell[2], J. Patrick Megonigal[3], James C. Stegen[2], and Ben Bond-Lamberty[1]

*Corresponding author, stephanie.pennington@pnnl.gov

1. Joint Global Change Research Institute, Pacific Northwest National Laboratory, 5825 University Research Ct. #3500, College Park, MD 20740 USA

2. Pacific Northwest National Laboratory, Biological Sciences Division, Richland, WA, USA

3. Smithsonian Environmental Research Center, Edgewater, MD, USA

Keywords: *spatial variability, soil respiration, temperate forest, carbon cycling*

**Abstract**

Soil respiration ($R_s$), the flow of $CO_2$ from the soil surface to the atmosphere, is one of the largest carbon fluxes in the terrestrial biosphere. The spatial variability of $R_s$ is both large and poorly understood, limiting our ability to robustly scale it in space. One factor in $R_s$ spatial variability is the autotrophic contribution from plant roots, but it is uncertain how the presence of plants affects the magnitude and temperature sensitivity of $R_S$. This study used one year of $R_s$ measurements to examine the effect of localized basal area on $R_S$ in the growing and dormant

seasons, as well as during moisture-limited times, in a temperate, coastal, deciduous forest in eastern Maryland, USA. In a linear mixed-effects model, tree basal area within a 5 m radius ($BA_5$) exerted a significant positive effect on the temperature sensitivity of soil respiration. Soil moisture was the dominant control on $R_S$ during the dry portions of the year while soil moisture, temperature, and $BA_5$ all exerted significant effects on $R_S$ in wetter periods. Our results suggest that autotrophic respiration is more sensitive to temperature than heterotrophic respiration at these sites, although we did not measure these source fluxes directly, and that soil respiration is highly moisture-sensitive, even in a record-rainfall year. The $R_S$ flux magnitudes (0.35-15.3 $\mu$mol m$^{-2}$ s$^{-1}$) and variability (coefficient of variability 10%-23% across plots) observed in this study were comparable to values observed in similar forests. Six $R_S$ observations would be required in order to estimate the mean across all study sites to within 50%, and 516 would be required in order to estimate it to within 5%, with 90% confidence.  A better understanding of the spatial interactions between plants and microbes, as well as the strength and speed of above- and belowground coupling, is necessary to link these processes with large scale soil-to-atmosphere C fluxes.

**Introduction**

Soil respiration ($R_s$), the flow of $CO_2$ from the soil to the atmosphere, is among the largest C fluxes in the terrestrial biosphere (Granier et al., 2000, Bond-Lamberty, 2018; Le Quéré et al., 2018), but remains poorly constrained both temporally and spatially at all scales. Unlike other large C fluxes such as net primary production, net ecosystem exchange, and gross primary production, Rs cannot be measured, even indirectly, at scales larger than a few square meters (Bond-Lamberty et al., 2016). Though global-scale Rs varies between vegetation types and

biomes (Raich et al., 2002; Raich and Schlesinger, 1992), and responds to disturbances such as land use and climate changes (Hursh et al., 2017; Schlesinger and Andrews, 2000), it is uncertain how these patterns arise from local-scale variability, limiting our ability to robustly scale the process.

[revised manuscript text omitted]

*Statistical analysis*

Respiration data were checked visually for artifacts or unusual outliers, but we did not exclude any data *a priori*. Data were then combined with the proximity measurements described above based on collar number. We used a linear mixed-effects model to test for the influence of $BA_5$ on $R_s$, treating temperature, soil moisture (SM), and $BA_5$ as fixed effects, and site as a random effect (Equation 1). $R_s$ frequently follows a nonlinear response in relation to SM, so a quadratic SM term (Sierra et al., 2015) was included in the model. 
[revised manuscript text omitted]

| | Power (1 - $\beta$) | | | | | |
|---|---|---|---|---|---|---|
| delta | 0.5 | 0.6 | 0.7 | 0.8 | 0.9 | 0.95 |
| 0.05 | 61 (24) | 95 (37) | 143 (55) | 219 (84) | 362 (138) | 516 (196) |
| 0.10 | 16 (6) | 24 (10) | 36 (14) | 55 (21) | 91 (35) | 129 (49) |
| 0.25 | 3 (1) | 4 (2) | 6 (3) | 9 (4) | 15 (6) | 21 (8) |
| 0.50 | 1 (1) | 1 (1) | 2 (1) | 3 (1) | 4 (2) | 6 (2) |

**Table 3.** Summary of linear mixed-effects model testing main hypothesis of the effect of nearby tree basal area on soil respiration (the dependent variable). Terms tested include soil temperature at 5 and 20 cm ($T_5$ and $T_{20}$ respectively), basal area (BA), and soil moisture (SM). Model AIC = 381.6, marginal $R^2 = 0.36$.

| | value | degrees of freedom | t-value | p-value |
|---|---|---|---|---|
| **(Intercept)** | $-0.767 \pm 0.148$ | 440 | -5.199 | 0.000 |
| **$T_5$** | $0.010 \pm 0.009$ | 440 | 1.055 | 0.292 |
| **$BA_5$** | $0.022 \pm 0.219$ | 440 | 0.098 | 0.922 |
| **$T_{20}$** | $0.095 \pm 0.011$ | 440 | 8.397 | 0.000 |
| **SM** | $2.505 \pm 0.699$ | 440 | 3.581 | 0.004 |
| **$I(SM^2)$** | $-3.542 \pm 1.144$ | 440 | -3.095 | 0.002 |
| **$T_5{:}BA_5$** | $0.079 \pm 0.036$ | 440 | 2.181 | 0.030 |
| **$BA_5{:}T_{20}$** | $-0.069 \pm 0.041$ | 440 | -1.689 | 0.092 |

**Figure 1** | a) Tree proximity measurement schematic. Distance to each tree was recorded within a 15 meter radius of each soil respiration measurement point, along with DBH and species. b) Map of the Smithsonian Environmental Research Center with the three sites labeled in black.

a)

[Figure]

b)

**Figure 2** | CO₂ flux over time from April 2018 to April 2019 for 36 measurement points across three sites; red line shows the seasonal trend using a loess smoother.

[Figure]

**Figure 3** | Cumulative basal area for each collar (N = 36) up to 15 meters; color indicates number of trees at each distance. Inset graph shows a close up of 0 to 5 meters for more detail.

[Figure]

[Figure]

**Figure 4.** Residuals of a soil respiration model, incorporating temperature and soil moisture as independent variables, versus cumulative tree basal area within 5 m, by site. Each point is an individual observation (cf. **Figure 2**). Regression lines are shown for each site; black line is the overall trend. Note that 5 extreme points are out of the plot but are accounted for in the regression lines.

[Figure]

**Figure 5.** Test of robustness of results, run at various distances from soil respiration measurement collars (x axis). Lines show the variable importance (calculated as $R^2$ partitioned by averaging over orders; see Methods) of basal area (BA), as well as the interaction of BA and temperatures at 5 and 20 cm ($T_5$ and $T_{20}$ respectively). Vertical dashed line shows the 5 m radius used in **Table 3** and **Figure 4** results. Note that 'missing' BA:T20 (in yellow) dots at distances < 5 m and >12 m mean that the terms were dropped from the model and are thus not significant.

[Figure]

--- END OF MANUSCRIPT ---